# “Targeting Design” of Nanoparticles in Tumor Therapy

**DOI:** 10.3390/pharmaceutics14091919

**Published:** 2022-09-11

**Authors:** Tingting Yang, Jingming Zhai, Dong Hu, Ruyue Yang, Guidan Wang, Yuanpei Li, Gaofeng Liang

**Affiliations:** 1School of Basic Medical Sciences, Henan University of Science & Technology, Luoyang 471023, China; 2Department of General Surgery, The First Affiliated Hospital, College of Clinical Medicine, Henan University of Science & Technology, Luoyang 471003, China

**Keywords:** nanoparticles, drug delivery, targeted transportation, cancer therapy, application

## Abstract

Tumor-targeted therapy based on nanoparticles is a popular research direction in the biomedical field. After decades of research and development, both the passive targeting ability of the inherent properties of NPs and the active targeting based on ligand receptor interaction have gained deeper understanding. Unfortunately, most targeted delivery strategies are still in the preclinical trial stage, so it is necessary to further study the biological fate of particles in vivo and the interaction mechanism with tumors. This article reviews different targeted delivery strategies based on NPs, and focuses on the physical and chemical properties of NPs (size, morphology, surface and intrinsic properties), ligands (binding number/force, activity and species) and receptors (endocytosis, distribution and recycling) and other factors that affect particle targeting. The limitations and solutions of these factors are further discussed, and a variety of new targeting schemes are introduced, hoping to provide guidance for future targeting design and achieve the purpose of rapid transformation of targeted particles into clinical application.

## 1. Introduction

According to the *2020*
*CA-A*
*Cancer*
*Journal*
*for*
*Clinicians*, there were 19.3 million new cancer cases worldwide and nearly 10 million deaths from cancer that year, indicating that cancer is still one of the biggest killers threatening people’s lives [1]. Cancer treatment includes surgical excision, radiation therapy and chemotherapy. Chemotherapy is the most commonly used treatment, but many chemotherapeutic agents have serious side effects including alopecia, bone marrow suppression and nephrotoxicity due to their short half-life and lack of targeting ability, and are prone to drug resistance. The emergence of a nanoparticle targeted drug delivery system provides a new breakthrough for tumor therapy.

Nanoparticles (NPs), as drug carriers, can deliver a variety of drug molecules, such as small molecule chemotherapeutic drugs, peptides, proteins and nucleic acids, to the desired target sites in a controlled manner [2,3,4,5,6]. Currently, a variety of nanocarriers, such as liposomes, dendritic macromolecules, exosomes, inorganic NPs, viral-like particles, protein and polypeptide NPs, have made certain progress in drug delivery [7,8,9,10,11,12]. Compared with administration alone, NP delivery systems have unique advantages in the treatment of cancer, such as extending the half-life of drugs to reduce side effects, improving the accumulation of drugs in tumors by passive or active targeting, or modifying the particle surface to promote cell uptake; they can even provide new application opportunities for powerful anticancer drugs that have been abandoned due to poor pharmacokinetics [2]. Initially, NPs were designed to deliver individual chemotherapeutic drugs and improve their solubility and pharmacokinetic characteristics. After decades of development, NPs have become a multifunctional targeted drug delivery platform [13], which is mainly reflected in the following aspects: (1) The types of commodity to be delivered include biological drugs such as proteins and nucleic acids; (2) Release mode is developed from single sustained release to environmental responsiveness release; (3) The mode of targeting changes from passive to active, or double targeting; (4) There is a change from single drug delivery to multi-delivery, giving full play to the synergy of drugs; (5) The targets range from tumor and tumor microenvironment to intracellular organelle targeting; (6) There are integrated multiple treatment methods, such as immunotherapy combined with chemotherapy, tumor detection combined with chemotherapy, and an integrated diagnosis and treatment platform is developed.

At the same time, the targeted delivery of NPs is also facing various challenges, and the corresponding targeted design or modification for various biological obstacles can produce real clinical therapeutic significance for tumors. After entering the body for several hours, particles will be cleared by the reticuloendothelial system (RES), mainly concentrated in the liver and spleen, or cleared by the kidney (5–8 nm) according to particle size, resulting in a significant decrease in the number of particles [14]. This first-pass effect is the first barrier to hinder the targeted delivery of particles, and it is necessary to consider how to escape the effect. After entering the body, the NPs are not directly recognized by macrophages, but first combine with various serum proteins through long-range electrostatic, van der Waals force and short-range hydrophobic interaction to form a “particle-protein” complex, also called “protein crown”, which changes the final biological characteristics of NPs [15]. Analysis of crown proteins revealed two types of proteins: opsonin and non-opsonin [16]. Opsonin adsorption enables NPs to be recognized by RES and quickly removed from circulation by first metabolism [15], while non-opsonin adsorption extends the circulation of NPs and eventually accumulates in tumor vessels, and then reaches surrounding tissues through vascular leakage or active endocytosis of the vessel wall. Extravasation of NPs from blood vessels into surrounding tumor tissues is also an extremely complex process affecting NPs delivery.

For tumors, their abnormal proliferation characteristics make the tumor vascular system highly heterogeneous in function and morphology [17]. Many irregular and disorganized vascular structures in tumors lead to uneven blood flow distribution, also known as heterogeneous blood flow, which makes many drug-loaded NPs not evenly dispersed in the entire tumor tissue, and some areas with poor blood flow are more prone to drug resistance [17]. In addition, the permeability of particle vessels also varies with the development stage of the tumor, and Jang found that the new vessels in the early stage of the tumor were more leaky than the mature vessels in the late stage [18]. At the same time, it is also necessary to consider whether the NPs can successfully infiltrate into tumor tissue, which is related to tumor interstitial pressure and the tumor extracellular matrix (ECM). This quantifiable pressure of fluid in the tumor stroma, which is related to the size of the tumor and the distance close to the center of the tumor, also hinders the delivery of NPs [19]. The interplay between the rapid proliferation of tumor cells, the high permeability of tumor blood vessels, and the lesser lymphatic drainage in the stroma all lead to increased tumor interstitial pressure [20]. The enhanced tumor interstitial pressure hinders the spread of drugs, so the areas with high interstitial pressure often lack the accumulation of drugs [21]. In terms of drug delivery, the rich and dense fibrous network of the ECM constitutes a powerful physical barrier that hinders the spread and distribution of particles within the tumor stroma [18]. Abnormally increased ECM can also compress blood vessels and lymphatic vessels, increase the interstitial fluid pressure in the tumor center, and further hinder the transport of drugs in the space. The final hurdle of NPs delivery is whether it can be successfully internalized by tumor cells. This internalization can be achieved passively or actively. With the study of tumor uptake mechanisms, it was found that the cellular internalization capacity was significantly correlated with the surface properties of NPs. For example, polyethylene glycol (PEG) enhanced the passive internalization of NPs and functionalized the surface of NPs, which further enhanced the active uptake of NPs by tumor cells [15].

Particle delivery is more difficult for brain tumors. The blood-brain barrier (BBB) is a selective permeability barrier, which is formed by the semipermeable boundary of the endothelial cells of the brain capillary wall where the end-feet of the astrocyte ensheathes the capillary, and the pericytes are embedded in the capillary basement membrane [22]. While this tight cellular junction protects the brain from toxins and pathogens, it also severely limits the entry of NPs from the bloodstream into the brain parenchyma. In addition, there are dynamic interfaces such as the blood-cerebrospinal fluid barrier and the blood brain tumor barrier, which selectively block the transport of substances. These brain barriers prevent almost all molecules from entering the brain from the blood to protect the central nervous system from toxic substances [23]. Therefore, for brain tumors, most drugs or drug-loaded NPs cannot cross these biological barriers to reach the tumor and exert antitumor effects. The protection mechanism of these organisms and the special physiological environment of tumors make it difficult for NPs to reach specific sites according to the expected route.

## 2. Effects of Physicochemical Properties of NPs on Targeting

### 2.1. Size

When NPs are used to deliver anticancer drugs, the physical and chemical properties of NPs, including size, charge, shape and internal or external properties, influence the final fate of NPs in the body [24]. Two cardinal factors should be considered when designing the size of particles: one is that they should be large enough to avoid being cleared by the kidney or invading into capillaries [25]; the other is that they should be small enough to escape the phagocytosis and clearance of the RES [26]. Studies have found that macromolecules larger than 40 KDa and NPs of 10~500 nm can leave the capillary bed and accumulate in the interstitial space of the tumor to achieve passive targeting [27]. This phenomenon of selective extravasation and retention in tumors is known as the enhanced permeability and retention (EPR) effect [27,28]. Dreher’s team found that particles with a diameter of hundreds of nanometers can effectively accumulate in target tissues. In the case of dextran, they found that increasing its molecular weight from 3.3 KDa to 2 MDa can reduce the permeability of dextran by two orders of magnitude. At the same time, it was observed that larger molecules were mainly accumulated on the surface of tumor blood vessels, while smaller molecules could penetrate deeper into tumor stroma and be more evenly distributed [29]. The clearance mechanism of NPs further reduces the effective particle size, and studies have found that particles larger than 200 nm are easier to be cleared by the body [30]. Secondly, the vasculature in the tumor is extremely permeable on account of the increase in the number of fenestrations and the size of the fenestrations combined with the influence of incomplete or abnormal basement membrane, which is an important way for NPs to accumulate and penetrate the tumor stroma [31]. The size of these fenestrations is usually 50~100 nm. Taking all these factors into account, effective nanocarriers should be in the diameter range of 10~150 nm, as shown in Figure 1, to prolong the particle cycle time and further increase particle accumulation in the tumor [32]. Although the EPR effect solves the dilemma that traditional chemotherapy drugs have no targeting ability in vivo, the number of NPs reaching the tumor site through the EPR effect still needs to be improved [25].

Therefore, a hierarchical targeting strategy to change the particle size was proposed [33]. Huang et al. developed a targeting system with small to large particle size, inspired by the protein assembly mechanism in nature. The small molecular structure peptides carrying indocyanine green (ICG) can effectively avoid the clearance of RES in the blood circulation, accumulate at the tumor site through the EPR effect, and self-assemble into macromolecular nanostructures by non-covalent bond force to prolong the retention time in tumor stroma [34]. The optical properties of the combined ICG molecules can distinguish normal tissue from cancerous tissue, greatly improving the accuracy of photothermal therapy. However, this strategy seriously affects the permeability of particles and the ability of internalization, which is not conducive to the delivery of intracellular target drugs. Therefore, variable size particles from large to small were developed for drug delivery. The original size of the NPs designed by this strategy can avoid the clearance of RES, and the disintegration of the original NPs under the stimulation of the tumor environment reduces the particle size significantly. Wang et al. prepared an NP capable of rapid size conversion at the tumor site [33]. The NPs were conjugated from Pt prodrug to poly (ethylene glycol)-block-poly (2-azepane ethyl methacrylate)-modified dendrimers of PAMAM dendrimers (PEG-b-PAEMA-PAMAM/Pt). At neutral pH, they can self-assemble into pH-sensitive cluster nano-monomer (SCNs/Pt), with a particle size of about 80 nm. In the acidic tumor environment (decreasing 0.1 to 0.2 pH units), PAEMA rapidly protonates in response to pH changes, and the particles instantly decompose into small particles less than 10 nm in diameter. The results show that this super-sensitive particle size decomposition strategy can effectively improve the tumor penetration and anti-tumor effect of the drug.

### 2.2. Shape

Another key factor in NPs design that affects pharmacokinetics and cellular uptake is the shape of the NPs. The shape design of NPs can play a pivotal role in circulation time, biodistribution, uptake by cells (Figure 2B) and targeting [25]. Among them, rod-shaped NPs emerge with higher absorption, followed by spherical, cylindrical and cubic NPs [35]. Aside from the traditional spherical NPs, viruses and bacteria in existing biological systems often use various shapes such as filaments or cylinders to evade clearance by the immune system [36]. Drawing lessons from the shape of viruses and bacteria, redesigning the shape of NPs reveals more profitable properties than spherical NPs. Some scholars have demonstrated that filamentous or wormlike micelles can circulate continuously in mice or rats for one week, providing more opportunities to interact with tumors and proving that the worm shape is more conducive to the uptake of NPs [37]. Through further investigation and analysis, it was found that for worm-like NPs, the strong push-pull effect of dynamic fluid flow can significantly extend their circulation time in the body. Secondly, multivalent contact between macrophages and worm-like NPs is needed to successfully swallow them, thus decreasing the particles’ clearance rate [35]. As the mechanisms of viral infection have become better understood, researchers have applied viral structure and function to the design of drug vectors. Yoo et al. created evenly distributed NPs of lipids, transferrin and DNA for gene delivery [38]. The NPs have a multi-center layered nuclear structure and a transferrin coating film, imitating the influenza virus and herpes virus with an envelope structure. Secondly, nanogel systems that mimic the structure and function of viruses have also been developed. They consist of a hydrophobic core containing the anticancer drug DOX and two layers of hydrophilic shells with tumor-targeted ligands. To simulate the capsid-like structure, PEG is attached to the core polymer as the inner shell, and bovine serum albumin is attached to the other end of the PEG as the outer shell. When the nanogel reaches the tumor site, its shape changes reversibly, releasing the drug and rapidly killing the tumor cells. The nanogel, on the other hand, moves to adjacent tumor cells and repeats the same process, similar to the infection cycle of a virus, to achieve the effect of continuing to kill tumors [39].

### 2.3. Surface Properties

The surface charge of NPs plays an important role in cell internalization (Figure 2A). The cell membrane is slightly negatively charged, so the positively charged particles are more likely to be absorbed into the membrane surface by cells through strong electrostatic adsorption [40]. The subsequent intracellular transport mechanism is also affected by the charge. The vesicles of positively charged particles formed by membrane invagination can fuse with lysosomes. Because the particles in the vesicles carry a large amount of positive charge, in order to maintain charge neutrality, many chloride ions remain in lysosomes, causing lysosome swelling and rupture [41]. The particles are released from lysosomes into the cytoplasm and exhibit perinuclear localization, highly killing tumor cells, whereas negatively charged and neutral NPs cannot elicit this effect and prefer to co-localize with lysosomes. On the biological level, due to the existence of a variety of proteins, lipids and sugars in the biological environment of such a serum, when positively charged particles enter the body, proteins are easy to adsorb on the surface of particles to form a “protein corona” [42]. This “protein corona” makes particles negatively charged on the whole, thus affecting cell internalization [43]. Secondly, positively charged particles are also more likely to bind to a variety of opsonins and be quickly cleared by macrophages, and the clearance rate increases with the increase of the absolute value of the particle surface charge [41]. Studies from animal models have shown that slightly positively charged NPs can accumulate in large quantities at tumor sites after systemic administration [44]. Therefore, particles with lower absolute charge find it easier to escape the removal of RES and prolong the blood circulation time of particles. This protein crown not only accelerates particle clearance but also impedes cellular uptake, adding additional complexity to the study of the mechanism of the interaction between NPs’ own properties and organisms.

NPs usually have comparatively large surface-to-volume ratios and their surface layer is the main medium of interaction with the biological environment [45]. Modifying the surface properties of NPs to make them have high stability and long cycle characteristics has wide research value. Studies have shown that particles with hydrophobic surfaces are more likely to bind opsonin to be phagocytosed and cleared by macrophages compared with hydrophilic NPs [46]. In recent years, Scarso et al. changed the hydrophobic properties of the particle surface by adding hydrophilic polymers, and they successfully presented the particles’ long-cycle characteristics [45]. Alexis modified the surface of NPs with PEG to reduce the immunogenicity of particles, endowed particles with “stealth characteristics” to protect them from opsonin adsorption, and reduced the clearance of NPs by changing the shape, density and length of the PEG chain [30]. A number of non-ionic hydrophilic polymers were developed as substitutes to PEG, such as poly(*N*-vinyl-2-pyrrolidone) (PVP), poly(amino acids) and dextran [41], to avoid adsorption. However, although this method effectively increases the accumulation of particles, unfortunately, the “invisible coating” of particles can greatly hinder their uptake by tumor cells. Therefore, further research is needed to develop suitable NPs with general surface properties.

### 2.4. Intrinsic Property of Particles

A variety of NPs drug delivery systems have been developed in recent years based on proteins, polysaccharides, inorganic NPs, exosomes, biomimetic NPs and virus-like particles [7,8,9,10,11,12]. These particles have different intrinsic properties which affect cell internalization ability. This demonstrates that under the circumstance of the same particle size, shape and surface properties, the internalization rate of single-walled carbon nanotubes by cells is 1000 times that of gold NPs, which may be determined by the inherent properties of carbon and gold [40].

Liposomes, as self-assembled colloidal vesicles, can not only encapsulate hydrophilic drugs and siRNAs in their hydrophilic cores, but also hydrophobic drugs in their hydrophobic membranes, and are widely used to deliver various drugs to treat cancer [22]. When the drug-loaded liposome reaches the target location, the drug is delivered to the cytoplasm through the lipid bilayer of the fusion membrane by using the principle of “similarity and compatibility” between the liposome and the cell membrane in structure and composition. Liposomes also have the ability to penetrate the BBB, which is related to the hydrophilic-hydrophobic balance of lipids. The β-amphetaminylated cationic lipid NPs developed by Saha et al. were non-cytotoxic and can cross the BBB through active transcytosis [47]. In addition, cationic and weakly basic molecules, due to their positive charge, can electrostatically interact with the negatively charged plasma membrane of endothelial cells to penetrate the barrier through adsorption-mediated endocytosis. Due to the high expression of specific receptors on the surface of brain capillary endothelial cells and neurons, such as nicotinic acetylcholine receptors (nAChRs) [22,47,48] and transferrin receptors (TfRs) [49], the cross-functional liposomes developed by utilizing the characteristics of liposomes and the advantages of ligand-receptor targeting have shown bright prospects. Saha et al. developed cationic amphiphilic liposomes containing NIC-1 and NIC-2 (binding to nAChRs across the BBB) loaded with a potent anticancer drug WP1066 for selective targeted therapy of brain tumors [48]. The results showed that the overall viability of mice bearing in situ glioblastoma was significantly improved.

The carrier can also use its own unique inherent characteristics to obtain the ability to actively target tumors (Figure 2C). Certain stem cells or immune cells with tumor homing and long circulation ability can be used as potential target vectors for drug delivery. By extracting the membranes of these cells as the “outer layer” to wrap the NPs surface, a series of biomimetic NPs drug delivery systems with good biocompatibility while inheriting the advantages of the original cells were developed. At present there are many biomimetic NPs based on platelets, macrophages, tumor cells and red blood cell membranes (RBC). For example, RBC-coated NPs designed by Villa et al. were compared with bare particles; membrane-coated NPs were found to observably protract the blood circulation time and reduce clearance of the immune system [50]. Hu constructed an active circulating tumor cells (CTC) targeting platelet drug delivery system that achieved tumor-targeted therapy by utilizing the interaction between platelets and p-selectin receptors on CTC [51]. Jing et al. used platelets to encapsulate melanin NPs and DOX, and modified RGD peptide to effectively inhibit the growth of drug-resistant tumor cells [52]. Guo took advantage of the natural tropism of macrophages to tumors to prepare membrane-coated drug-loaded NPs for the treatment of ovarian cancer, showing a strong therapeutic effect [53]. Furthermore, it was found that the chemokine receptors CCR2 and CCR4 of biomimetic macrophage membrane NPs were highly expressed, which further enhanced the chemotaxis of macrophages and improved drug accumulation in tumor sites. Exosomes contain a variety of marker proteins such as CD9, CD63, CD81 and TSG101 on the surface, which can specifically bind to recipient cells and have attracted extensive attention as targeting vectors [54,55]. Researchers add targeted molecules to the surface of exosomes, or modify the source cells through genetic engineering to make their surface highly express tumor-targeted proteins or polysaccharides to obtain engineered exosomes, and thus design an active targeting system based on exosomes. Kumar et al. used folic acid functionalized bovine milk exosomes to deliver paclitaxel (PTX) and 5-fluorouracil (5-Fu) for targeted therapy of breast cancer, and showed that drug-loaded exosomes can significantly reduce drug side effects and improve their efficacy for breast cancer [56,57]. Kamerkar used exosomes from normal fibroblast-like mesenchymal cells to carry short interfering RNAs or short hairpin RNAs specific for oncogenic KrasG12D, a common mutation in pancreatic cancer. In a mouse pancreatic cancer model, this engineered exosome therapy suppressed tumors and significantly improved overall survival, demonstrating the targeting advantages of engineered exosomes [58]. Liang et al. obtained engineered exosomes by fusing CD63 transmembrane protein and APO-A1 sequence gene with genetic engineering technology, whose surface specifically expressed antibodies against tumor biomarkers to further enhance the cell targeting ability of exosomes [59].

In recent years, biological macromolecules such as polysaccharides, proteins and peptides with their own targeting ability have been developed as drug nanocarriers. Hyaluronic acid (HA), as a natural linear polysaccharide, has multiple functional loci and intrinsic affinity for CD44 [60]. By taking advantage of the overexpression of CD44 on the surface of cancer cells, Li et al. developed therapeutic NPs of hyaluronic-acid-coated magnetic polydopamine coupled with methotrexate, and proved that the prepared HA-NPs could accumulate in tumor sites in large numbers through magnetic resonance imaging [61]. Another advantage of HA is that it can be degraded by hyaluronidase 1(Hyal-1), which is highly expressed in a variety of malignant tumor cells, destroying the spatial structure of HA-NPs to release the encapsulated drugs. In Yoo’s study, the anticancer drug camptothecin (CPT) was loaded into HA-NPs, and in the presence of Hyal-1, the HA-NPs loaded with CPT degraded and discharged CPT rapidly. The results revealed a conspicuous increase in cytotoxicity of CPT-HA-NPs in a dose-dependent way compared to free drugs [62].

## 3. Improved Targeting Based on Ligand-Receptor Interactions

Compared with normal tissue, tumor tissue has the characteristics of acidic environment and some receptor overexpression, which provides research directions for active targeting. By modifying targeting ligands on the surface of particles and encapsulating or conjugating drug molecules to maximize drug delivery to target tissues or cells, high-efficiency and low-toxicity therapeutic effects are achieved. Excluding the influence of the physical and chemical properties of the particle itself, we only discuss the influence of the ligand-receptor-based strategy on the particle targeting ability and its limitations in the body.

### 3.1. Ligand-Receptor Binding Force

The targeting ability of NPs is affected by ligand affinity. High-affinity ligands can bind closely to the corresponding receptors so that NPs can be effectively attached to the membrane surface and endocytosed by cells. However, this high-affinity ligand cannot clearly distinguish tissues with different levels of receptor expression, causing unnecessary cytotoxicity to normal cells (Figure 3C). So, how can the ligand on the NPs effectively bind to the tumor cell receptors and avoid the normal cell receptors? The researchers cleverly solved this problem by exploiting the multivalence of ligand binding. “Multivalent interaction” is a situation in which a ligand can bind to multiple receptors at the same time in a biological system, showing the “super-selective” feature of the ligand to achieve the targeting of specific cells [63]. This property has been widely seen in nature, as in the attachment of viruses or bacterial pathogens to cells [64].

In order to clearly define the interaction between targeted particles and cells, some scholars use a computer to simulate the binding state between the targeted ligand of particles and cell surface receptors. Martinez-Veracoechea et al. designed a model in which NPs have separate spatial locations and do not compete for the same receptor to investigate monovalent and multivalent NPs’ targeting selectivity. The single bond of the multivalent NPs in the model is 5 KT weaker than the monovalent ones, but can bind 10 receptors simultaneously (KT represents the binding strength of the ligand-receptor monovalent, where K is the Boltzmann’s constant and T is the absolute temperature). The results showed that the binding selectivity of monovalent NPs was very small. When the receptor concentration was tripled, the average number of binding particles only increased from 5.4 to 9.7, and changed linearly with the increase of receptor concentration. While under the same conditions, the average number of binding particles of multivalent NPs increased by nearly 10 times, and the binding receptors increase far exceeded the linear change [65]. This “super-selective” phenomenon revealed the targeting ability of multivalent NPs. On the cell surface where the receptor concentration reached a certain threshold, the binding force between the particle and the receptor was significantly enhanced, laying the foundation for the next step of endocytosis.

Hong et al. attached 2~14 folic acid and AlexaFluor488 dye (AF488) to the G5 dendritic molecular scaffold and synthesized FAR-targeting nanocomposite (G5-Ac-AF488-FAx). Surface plasmonic resonance was used to quantitatively determine the KD value of the interaction between the nanocomposite and the folate-binding protein on the cell surface. The results showed that compared with free folate molecules, the binding affinity of the nanocomposites increased 2500–170,000 times, which significantly enhanced the residence time of multivalent NPs on the target cells, and promoted the uptake of particles by cells [66]. Carlson synthesized a bifunctional conjugate and artificially created a multivalent binding platform on the membrane surface by non-covalent interaction. One end of the bifunctional conjugate can bind with integrin receptors on the membrane surface with high affinity, and the other end was a single-Gal epitope (Kd ≈ 10 µM) that was weakly bound to anti-Gal antibodies. The constant presence of -Gal antigens in the body ensured the availability of circulating anti-Gal antibodies (IgM or IgG) and complement. Due to the low expression of integrin receptors in normal cells, the antibody can only bind in a monovalent manner to the bifunctional conjugate, which was not enough to attach to the cell surface, while for tumor cells with high expression of integrin receptors, multivalent interactions were utilized, which can effectively bind IgM to cells to activate the complement cascade and mediate the lysis of target cells [67]. This targeted design using “multivalent binding” has a certain guiding role in tumor therapy.

### 3.2. Activity Maintenance of Targeted Ligands

Another factor that affects targeting is the activity of the ligand. When the ligand is modified by chemical coupling or electrostatic adsorption, the activity of the ligand is often affected, resulting in an “off-target effect” or other side effects. For example, when fibrinogen was attached to the surface of polyacrylic-acid-coated gold NPs, it was denatured and bound to the integrin receptor MAC-1, leaded to inflammation [40].

A variety of proteins in biological liquids are in a relatively stable equilibrium state [68], and the entry of NPs destroys this equilibrium state in the body. Many biomolecules can quickly adhere to form protein shells with NPs as scaffolds through a variety of mechanisms, giving NPs a new biomolecular interface. A study of particle surface proteins revealed that there were significant limits to the number and type of proteins adsorbed (usually only a few hundred), and that the composition of the crowns also changed over time, but the total amount of adsorbed proteins remained relatively constant. Some high-affinity proteins preferentially bind to particles to form a “hard corona” as an inner layer, which exchanges with the external medium on a time scale of hours [69]. The “soft corona” in the outer layer is linked to the hard corona through weak protein-protein interactions and is exchanged with the external medium on a time scale of seconds or minutes [70]. This ongoing adsorption and desorption behavior of the protein corona, controlled by the “Vroman effect”, correlates with the affinity of the protein [15]. These adsorption proteins regulate the fate of particles, such as the adsorption of IgG, IgA or complement proteins (C3b, C4b), which enables the NPs to be quickly cleared by phagocytes [71]. Secondly, NPs are often highly accumulated in the liver and spleen with a high concentration of 30–40%, while only 1–2% can be achieved in other organs, including tumor tissues. This strong liver and spleen localization ability of NPs may also be dominated by non-specific protein adsorption [72].

It is worth mentioning that this adsorption phenomenon can also change the structure and spatial position of the ligand, hinder ligand binding or directly mask the ligand and make it lose its targeting ability (Figure 3B). So, how does one ensure the activity of the targeted ligand? Previous studies have shown that the intrinsic properties of ligands, the composition of linkers, and the conformation and epitope presentation of ligands or other details affect ligand activity. The non-specific interactions between various biological macromolecules and nanomaterials, such as non-covalent binding caused by amino acid side chains, can significantly change the ligand structure, so the influence of NPs on ligand activity should also be considered. Studies have shown that adjusting or modifying the size [73], hydrophobic parameters [74] and charge properties [75] of nanomaterials can effectively maintain the primary structure of targeted ligands. Some scholars have even proposed that under some conditions, NPs can in turn protect the original structure of targeted ligands and maintain protein activity [76].

The use of crosslinking agents to bind ligands on the surface of particles in an irreversible manner by covalent coupling is a very promising scheme. The selection of crosslinking agents should consider whether the available functional groups carried by the particle and the targeted ligand can form stable covalent bonds under appropriate reaction conditions. The common crosslinking agent is carbodiimide, which can dehydrate and condense the carboxyl group on the surface of the carrier with the amino group carried by the ligand to form an amide bond. In this reaction, the carboxylate particles first react with carbodiimide to form an active intermediate, and then react with the amino group on the ligand to form a stable covalent binding compound. This chemical coupling method can be completed in one step and has a certain level of biocompatibility. At present, a variety of NPs have been successfully combined with targeted ligands using this method, such as silicon [77], gold [78] and other NPs. *N*-hydroxysuccinimide or sulfo-*N*-hydroxysuccinimide is used to form active esters between aptamers and substrates or drug delivery carriers to produce stable amide contraction [79]. This covalent coupling has been successfully applied to gold [80] and silicon [81] NPs by succinimide ester-amine chemistry or maleimide–thiol chemistry. For amino-functionalized NPs, bifunctional glutaraldehyde is usually selected as the crosslinking agent between NPs and targeted protein amino groups, and the reversible imine groups formed can be reduced to fixed secondary amines [82]. Thiol-maleimide coupling utilizing primary amines on protein surfaces is also a highly specific covalent coupling strategy [82]. The Huisgen cycloaddition “Click” reaction coupling method offers significant improvements in the range and efficiency of binding. Among them, NPs can be specifically bound by copper-free azide-alkyne cycloaddition (an efficient bio-orthogonal reaction) with the introduction of azide or alkyne groups on the surface of biomacromolecules, which is considered to be particularly attractive [83]. Secondly, biomolecules can also be linked by non-covalent strategies, including affinity interaction, i.e., streptavidin-biotin interaction, or metal coordination, i.e., polyhistidine tag and Ni^+2^ chelates with immobilized nitrilotriacetic acid [79]. These covalent and non-covalent strategies have been used to immobilize a wide range of biomolecules, including proteins, enzymes, peptides and nucleic acids on the surface of NPs. However, this binding mode often leads to the simultaneous attachment of multiple ligand sites, which leads to changes in the structure of ligands and affects the active expression of ligands [84]. For the target protein, the ligand must present the active site in the correct way to recognize and bind the receptor, so the actual site of ligand coupling should also be considered [62]. In order to obtain correct epitope presentation, some scholars have studied the sites of ligand coupling, combined negatively charged gold NPs with different sites of cytochrome C through covalent mercaptan gold bonds, and studied their effects on protein structure by circular dichroism [85]. The results showed that the binding of particles on the *N*-terminal and C-terminal folding surfaces of the core structure of cytochrome C had the most serious effect on the protein structure, indicating that the actual binding sites of particles and ligands can greatly affect the targeting function of ligands.

Taking advantage of the competitive adsorption of particles by proteins with high affinity, using some proteins with strong affinity and localization function in vivo as the target ligand can be regarded as a new method to improve the targeting of NPs. Caracciolo et al. have synthesized lipid NPs using 1, 2-dioleoyl-3-trimethylammonium propane (DOTAP) and DNA in this way. Nano liquid chromatography tandem mass spectrometry (Nano LC−MS/MS) was used for qualitative and quantitative analysis of adsorbed proteins on the surface of particles [86]. Combined with gene ontology to search for corresponding receptors, it was found that vitronectin is a very promising active targeting ligand protein, which can recognize the highly expressed α_ν_β_3_ integrin receptor on the surface of tumor cells. Cell results showed that MDA-MB-435S cells with high expression of the α_ν_β_3_ integrin receptor showed a 5-fold increase in particle uptake compared to normal cells. This method can effectively avoid the deactivation of ligand.

### 3.3. Types of Bioactive Ligands

Targeted ligands for drug delivery systems typically include proteins and polysaccharides, artificial peptides, aptamers and small molecules (Figure 3A). These ligands have different targets and enter cells through different endocytosis pathways, which have shown good effects in anticancer therapy. Table 1 summarizes and collates some widely reported studies based on ligand receptor targeting in recent years.

Human transferrin (Tf) is a biodegradable and non-immunogenic iron binding protein, and its receptor (TfR) is highly expressed in a variety of tumor tissues [49]. A variety of drug delivery systems using Tf as targeted ligand have been developed and have achieved certain results [87]. Lam synthesized a transferrin-functionalized NP (Tf-NP) for targeted delivery to the brain [88]. In vivo imaging showed that Tf helped NPs cross the BBB and showed specific accumulation in brain tissue. Monoclonal antibodies (mAbs) specifically target antigens or receptors that are overexpressed in tumors and produce antitumor effects by blocking ligand-receptor binding or by inhibiting the expression of certain tumor-associated proteins [89]. Liszbinski prepared 5-Fu-coated gold NPs (AuNPs) and modified them with epidermal growth factor receptor mAbs for targeting colorectal cancer cells [90]. Flow cytometry showed that the mAbs successfully delivered drug-loaded NPs to cancer tissues and induced apoptosis/necrosis of colon cancer cells. Unlike proteins, polypeptides do not have high-level structures, and combinatorial phage libraries can be used to screen polypeptide molecules with potential high affinity to tumor targets and synthesize them artificially by chemical methods [91]. Shao et al. used arginine-glycine-aspartate acid (RGD) as a targeted peptide to recognize the α_v_β_3_ integrin receptor highly expressed in gastric cancer and developed a multifunctional NP targeted drug delivery device [92]. Compared with non-targeted NPs, multifunctional NPs clearly represent the tumor location and edge, demonstrating the powerful tumor targeting ability of RGD peptide. Similarly, cyclic RGD ligand was employed to target α_v_β_3_ and α_v_β_3_ integrin receptors that are abundantly expressed on vascular endothelial cells of human pancreatic adenocarcinoma for systemic anti-angiogenic therapy [93,94]. Aptamers are composed of single-stranded DNA or RNA oligonucleotides that can fold into a stable conformation. At present, suitable aptamer nucleic acid sequences can be found from random nucleic acid libraries by high-throughput screening methods. The synthesized aptamer is an ideal targeting ligand due to its small size, high sensitivity, biodegradation and non-immunogenicity [95]. Dhar’s group selected prostate-specific membrane antigen (PSMA) as a tumor target, and conjugated PSMA aptamers on the surface of cisplatin-loaded NPs prepared by poly(D,L-lactic acid-*co*-glycolic acid) and poly(ethylene glycol) chains to kill prostate cancer cells [96]. Experimental results showed that the drug-carrying NPs using PSMA aptamer could accumulate in large numbers in prostate cancer cells, and the killing rate of prostate cancer cells was 80 times higher than that of free cisplatin. Small-molecule folate acid (FA) is an essential vitamin for cell life and a highly selective tumor marker [97]. Folate-modified NPs can release cytotoxic drugs into the cytoplasm of tumor cells to induce apoptosis through folate-receptor-mediated endocytosis [98]. Liang developed a novel photosensitizer FA-TiO_2_-Pc by coupling nano-titanium dioxide (TiO_2_) particles with the tumor-targeting agent folic acid (FA) and photosensitizer phthalocyanine (Pc) [99]. When the mice were treated with low doses of FA-TiO_2_-Pc and exposed to low light, tumor growth was inhibited and side effects were significantly reduced compared to the untargeted group.

### 3.4. Receptor-Mediated Endocytosis

Different cells have different internalization mechanisms. Understanding the internalization mechanism of particles has great guiding significance for improving particle targeting. The original accumulation of particles in tumor sites mainly depends on the EPR effect, while receptor targeting is more reflected in the ability of cells to absorb particles. The ligand-modified NPs bind to specific receptors and enter cells through receptor-mediated endocytosis, which is a rapid ligand-targeted internalization method. However, when the particles bind to the receptor, it does not mean that they can be successfully internalized into the cell, as follows. (1) The off-target after binding often occurs when the affinity of ligands is low or the monovalent binding force between ligands and membrane receptors is weak. At present, the polyvalent effect can significantly increase the binding force of the whole complex and facilitate endocytosis. (2) Particle efflux after internalization is mostly related to cellular multidrug resistance proteins, such as p-glycoprotein, multidrug-resistance-associated protein, breast cancer resistance protein and other efflux transporters that are highly expressed in most tumor tissues [105]. The ligand-receptor-mediated endocytosis forms vesicle-coated particles through membrane invagination, which can avoid the influence of these transporters and successfully deliver drugs into the cells. (3) After the particle is internalized, it escapes from the vesicle and enters the cytoplasm. After the ligand on the particle surface binds to the membrane receptor, the local Gibbs free energy decreases, which promotes the invagination of the membrane to form a closed vesicle that wraps around the particle. Subsequently, multiple vesicles form endosomes and fuse with lysosomes, releasing particles or delivering them to other organelles [106].

There was also a significant correlation between the receptor-mediated uptake rate and particle size. Under the condition of a sufficient number of receptors, larger particles have larger ligand-receptor binding numbers than small particles, which can generate enough free energy to drive the membrane invagination. Of course, it is not the case that bigger particles are better. From a thermodynamic point of view, NPs with a particle size of 30~50 nm can bind to enough receptors and effectively attach to membrane receptors to drive membrane internalization [107]. However, particles above 50 nm bind to a large number of receptors on the membrane surface, resulting in excessive local consumption of receptors and limiting the binding of subsequent particles [108]. For particles smaller than 30 nm, they can bind to membrane receptors, but the number of receptors is too small to drive the membrane invagination.

Different cells have different internalization mechanisms and their uptake of particles is also different [109]. Most cells rely on clathrin for endocytosis, such as clathrin-dependent asialoglycoprotein (ASGP) and its receptor (ASGPR)-mediated endocytosis, where ASGP and receptor binding are internalized in clathrin pits [110]. In addition, there are many other endocytosis mechanisms, such as caveolin co-receptor endocytosis, liquid-phase endocytosis, lipid raft endocytosis or macroscopic endocytosis. Here, we mainly introduce receptor-mediated endocytosis. At present, a variety of high-affinity receptors have been used for targeted drug delivery, such as transferrin receptor [111], ανβ3 integrin receptor [112], CD44 receptor [60], epidermal growth factor receptor [90] and other receptor targets. When the ligand modified on the particle surface binds to the receptor target, the ligand and receptor complex move towards the pit of the membrane and accumulate with the help of the poly-subunit protein complex and adaptor protein, and then the membrane invaginates into the cell [113]. In subsequent processes, ligand receptor complexes in vesicles co-locate with lysosomes to degrade or escape lysosomes to target other subcellular structures. The mechanism of cell internalization is quite complex and is still not fully understood, but receptor-mediated endocytosis of drug-carrying particles is of great significance for tumor-targeted therapy, especially for some drugs acting on organelles.

### 3.5. Distribution and Recycling of Receptors

The cell membrane is a dynamic biofilm, and the location of the receptor on the membrane is not constant. Due to different cell types and functions, the expression of various proteins on the membrane surface is also significantly different. The distribution and density of receptors, recycling ability and mobility of receptors on the membrane affect the targeting of NPs and the uptake ability of cells. For optimal endocytosis, the density of receptors must be much greater than the number of ligands. However, in practice, receptors may change under the influence of some factors; for example, human folic acid receptor (hFR), as a receptor mediating the intracellular transport of folic acid and methotrexate (MTX), was downregulated by 1.8–24 times in acquired MTX-resistant human nasopharyngeal epidermoid carcinoma (KB) cells [114].

Secondly, receptor-mediated endocytosis also does not occur indefinitely, and when receptor-ligand complexes enter the cell, it results in a significant decrease in the number of receptors on the membrane surface, thus limiting the binding of other subsequent particles [115]. Therefore, whether receptor-mediated endocytosis can continue to occur is closely related to the recycling capacity of the receptors. Paulos et al. quantified the recycling rate of folate receptors (FR) using folate-related radiographic imaging agents [116]. It was found that the circulation rate of FR varied greatly among different cancer tissues, ranging from 6 to 20 h. Studies have shown that optimal endocytosis efficiency is achieved when the dosing frequency of particles is slightly lower than the circulating frequency of tumor FR [116]. This receptor-mediated rapid and repetitive endocytosis behavior can reduce the toxic effect of particles on other tissues. At present, the specific mechanism of receptor circulation is not fully understood. It is generally believed that the ligand-receptor complex is internalized to form the early endosome and then acidifies. Under acidic conditions, the receptor-ligand dissociates, and the receptor buds into separate vesicles to return to the cell surface to achieve the purpose of receptor recycling (Figure 3C). Studies have shown that ethanol exposure can cause changes in the distribution of receptors in the endosomes and affect receptor recycling. Dalton studied the distribution of ASGP receptors in “early” and “late” endosomes and found that, compared with the control group, there were more ligands binding to receptors in the endosomes in the ethanol exposed group, indicating that ethanol exposure damaged the normal transport of ligands and receptors in the liver endosome, delayed the dissociation of receptor-ligand complexes, and finally, resulted in the inability of receptors to enter the circulatory pathway in a timely manner [117].

The expression of membrane receptors is not invariable under the action of external or internal factors [83]. So, the question arises: how to solve the problem of receptor expression variation? Artificial receptors were introduced into our field by developing artificial receptors to ensure adequate receptor-mediated internalization while avoiding nonspecific uptake in normal tissues. The introduction of safe and stable synthetic groups on the surface of tumor cells can act as artificial receptors. Nethi et al. engineered bone marrow mesenchymal stem cells and used *N*-azidoacetylmannosamine-tetraacylated to express azido-salicylic acid on the membrane surface. Among them, engineered NPs can specifically bind to azido-salicylic acid via copper-free azide-alkyne cycloaddition, an efficient biorthogonal reaction [83]. The results showed that the functionalized NPs could effectively recognize the binding azide groups on the surface of MSC, and the retention time of the NPs in the tumor site was extended to 2 weeks. Animal experiments showed that this two-step targeting strategy effectively inhibited tumor growth and prolonged survival in mice. This artificial receptor method avoids the limitation of the instability and number of natural receptors and has a unique advantage in controlling receptor concentration and ligand affinity. However, this strategy only increases particle adhesion on the membrane surface, and there is no clear evidence that artificial receptors can effectively promote particle endocytosis. Considering the intracellular fate of the binding particles, the artificial receptor targeting strategy needs to be further optimized and designed to introduce more “flexible” targeting receptors on the membrane surface.

### 3.6. Amplified Receptor Signal

Traditional targeting strategies are constructed based on the EPR effect of NPs and ligand-mediated active targeting. One must change the traditional targeting thinking mode, transfer the field of vision from modified particles to receptor modification, and increase the accumulation of particles by amplifying the “receptor signal” at the tumor site, which has become a new breakthrough in tumor-targeted therapy. At present, some scholars have designed a targeted delivery strategy for signal amplification based on this method. Endogenous multi-step biological cascade reaction with strong signal amplification is selected as the signal module, and the signal is effectively dispersed to recruit circulating drug-carrying particles to gather at the tumor site. Maltzahn selected the ubiquitous coagulation cascade in plasma as a tumor signal amplifier [118]. Gold nanorods (NRs) were used to accurately set the signal module. After 24 h of NRs injection, obvious local accumulation of fibrinogen (a protein labeled with near-infrared fluorescent dye and used to evaluate tumor temperature) occurred at the tumor site, indicating that NRs fever destroyed tumor blood vessels and successfully activated the coagulation cascade under near-infrared light irradiation. The targeting of the drug-loaded NPs was studied, and it was found that the accumulation of DOX liposomes in the tumor site was enhanced by nearly 40 times in the heating group, which greatly improved the anti-tumor effect. This double NPs targeted drug delivery system, which amplifies the disease signal in vivo, can significantly improve the specific accumulation of NPs at the tumor site and avoid the effects caused by the variation or downregulation of receptor expression on the tumor membrane. It has become a novel and promising new method in the field of tumor therapy, but this strategy is not applicable to organelle targeted drugs.

## 4. Stimulus-Responsive Targeting Strategies

Based on the unique characteristics of the tumor microenvironment, NPs are designed as an intelligent responsive drug delivery system, which can remain “inert” in normal tissues, but can change its state to trigger local drug release under certain environmental stimuli at the tumor site, and improve drug concentration to enhance the drug treatment effect (Figure 4). Currently, environmentally responsive drug delivery systems that have been developed include physical response (temperature, light, magnetic, ultrasonic, etc.), chemical response (pH, reduction) and biological response (enzyme, glucose) [119]. Table 2 summarizes some relevant studies in recent years.

### 4.1. Physical Response (Temperature, Magnetism, Light)

Compared with normal tissue (37 °C), temperature (hyperthermia) in the range of 40~45 °C can increase blood flow in tumors and improve vascular permeability [120]. Therefore, with the increase of temperature, a large number of heat-sensitive NPs can accumulate in the hyperthermia area of the tumor and undergo phase transition and conformational changes [121]. Dipalmitoyl-phosphatidyl-choline (DPPC) is a thermosensitive liposome with an appropriate phase transition temperature, with a gel-liquid crystalline transition temperature (Tg) of 41 °C. Peng’s group designed a thermo-sensitive magnetic cationic liposome (TSMCL) drug delivery system [122]. The DOX and SATB1-shRNA were loaded into liposomes, and the gene delivery efficiency of TSMCL was determined by in vitro experiments. The results showed that DOX in the liposome had heat-sensitive release property and successfully silenced the SATB1 gene in cells, showing a good targeted therapeutic effect.

Magnetic targeted NPs delivery systems usually contain therapeutic payloads and magnetic active components [123]. These NPs can be controlled by externally applied magnets or alternating magnetic fields to concentrate on the pathological site, and at the same time control the drug release from the carrier to achieve the purpose of targeted drug delivery and magnetic hyperthermia [124]. Dilnawaz’s group used aqueous glycerol monooleate as a coating to coat magnetic NPs to prepare magnetic NPs (GMO-MNPs) [125]. GMO-MNPs can be loaded with different anticancer drugs (paclitaxel or rapamycin) with an encapsulation efficiency of about 95%. The killing effect of GMO-MNPs on tumor cells was significantly enhanced under the effect of the external magnetic field.

Light-sensitive nanocarriers can initiate remote spatial-temporal control of the release of drugs under the action of light of specific wavelengths and provide an attractive choice for targeted drug delivery [126]. The design of photosensitive NPs usually encapsulates the photo-responsive groups or chromophores in the particles or couplings on the surface of the particles. Various photoreaction mechanisms such as photo-isomerization, photo-crosslinking or photo-degradation of the polymerization main chain are used to induce the release of drugs in the NPs [127]. Azobenzene and its derivatives are splendid photo-responsive groups, which can transform non-polar trans isomers into polar cis isomers under UV irradiation, and change the polarity of nano-carriers. Subsequently, under the action of visible light, they are transformed into trans isomers to reduce the stability of the nanocarrier and release the encapsulated drug [128]. Secondly, drug release from photosensitive NPs may be a one-off or repeatable “switch” [129]. For example, photo-polymerization or photo-crosslinking involves a polymerizable double bond. Under direct irradiation of light, the double bond of the hydrophobic part of the nanocarrier bilayer polymerizes, resulting in the shrinkage of the bilayer part to create pores and repetitive “switches” to release the payload [127].

### 4.2. Chemical Response (pH, Reduction)

Due to different metabolism, there is a certain pH gradient between tumor and normal tissue [130]. A variety of pH-sensitive NPs have been developed by using pH-sensitive groups such as carboxyl and amino groups [131] and chemical bonds such as hydrazone, acetal, ester and coordination bonds [132] to achieve drug delivery at cancer tissue sites. Dong et al. efficiently loaded the photosensitizers Ce6(Mn) and DOX with nanoscale calcium carbonate and modified them with PEG to prepare multifunctional NPs (CaCO_3_@Ce6(Mn)-PEG(DOX)) [133]. Real-time monitoring of drug release using magnetic resonance imaging (MRI) revealed that NPs at the tumor site exhibited enhanced T1 signals. These results indicated that the complex particles were highly sensitive to pH and would rapidly degrade and release the drug Ce6(Mn) in a slightly acidic environment. Glucose oxidase (GOD)-loaded polymersome-based nanoreactors have also been developed to selectively catalyze glucose to produce hydrogen peroxide in the acidic pH of the tumor (pH ~6.4) but not in the normal healthy tissues. Hydrogen peroxide increases the oxidative stress of cancer cells and destroys phenylboronic ester to release quinone methide, which depletes the glutathione level, thereby suppressing the antioxidant activity. Ultimately, the nanoreactor efficiently kills the cancer cells via the synergistic effect of hydrogen peroxide and quinone methide [134]. Glutathione (GSH) is a tripeptide consisting of glutamic acid, cysteine and glycine containing γ-amide bond and sulfhydryl group and has two forms: reductive and oxidative in vivo [135]. The concentration of GSH in normal tissue is about 2–20 µM lower, while the concentration of GSH in the tumor site can reach 1–10 mM [136]. He et al. synthesized a novel dextran-Pt (IV) conjugate by attaching Pt (IV) to the side chain of hydrophilic dextran and encapsulating DOX [137]. Pt (IV) is not only a key component of vector composition, but also an antitumor prodrug. Under the strong reducing environment of the tumor, Pt (IV) was reduced to active Pt (II) and detached from the side chain of dextran, so that the carrier structure was destroyed and DOX was released. In vitro cytotoxicity demonstrated that the release of DOX and Pt (IV) can be triggered simultaneously under the control of a single reduction reaction, synergistically enhancing the antitumor effect.

### 4.3. Biological Response (Enzyme)

Common enzymes include matrix metalloproteinases (MMPs), gelatinases, hyaluronidases, esterases and phospholipase A2, which are dysregulated in cancer and have been successfully used in the design of enzyme-triggered NP delivery systems [138]. These enzymes are also considered to be biomarkers for cancer diagnosis and prognosis. Under the action of these enzymes, the ester or the designed short peptide sequence in the carrier can be cleaved to change the structure or surface properties of the carrier, and the drug can be released from the vector [139]. Qin et al. designed an enzyme-triggered drug delivery system based on the MMP2 enzyme [140]. DOX was chemically coupled with poly (ethylenimine)-co-poly (ethylene glycol) (PEI-PEG)-modified grafted GO via a PLGLAG peptide chain, which could be cleaved by MMP2. Under normal physiological conditions, the effect of DOX was limited, and its immanent fluorescence was quenched by GO. However, in the tumor site, the PLGLAG peptide chain was cleaved, DOX was released and its fluorescence characteristics were restored simultaneously, which were successfully used in orthotopic tumor imaging and tumor treatment. Kataoka et al. constructed an MMP transformable polymersome for co-delivering colchicine and marimastat to eliminate minimal relapsable cancer [141]. Han developed an enzyme-responsive charge-inversion nanoparticle (O-NP) consisting of hydrophobic oleic acid, MMP9 cleavable peptide and a glutamate-rich fragment [9]. Under the action of MMP9 in the tumor microenvironment, the surface charge of O-NP was successfully reversed from negative to positive. The results of the mouse model showed that O-NP was preferentially accumulated and endocytosed in MMP9-expressing tumors, which began to exert anti-tumor effects and effectively reduce the toxicity of normal tissues.

## 5. Conclusions and Future Perspectives

With the further study of targeted NPs and the in-depth research on targeted NPs, a variety of different NP drugs have been approved by the US Food and Drug Administration (FDA) or are in clinical trials. In 2005, Abraxane (albumin-bound paclitaxel) was the first FDA-approved passive-tumor-targeted polymeric nanomedicine for breast cancer treatment [13]. Subsequently, Myocet (liposomal doxorubicin), Daunoxome (liposomal daunorubicin), Onco-TCS (liposomal vincristine) and Oncaspar (PEG-L-asparaginase) have all been approved for targeted therapy in different cancers [13]. Many other nanomedicines have also been in the clinical evaluation stage, such as liposomes loaded with bupivacaine for the treatment of malignant female reproductive system tumors, and have reached phase IV clinical trials; nab-paclitaxel for the treatment of squamous non-small cell lung cancer has also reached phase IV; pegylated liposomes loaded with doxorubicin for borderline ovarian serous tumors has now reached phase III clinical trials, etc. [14]. As more nanomedicines were developed and entered clinical trials, it was found that while they were able to prolong circulation time and alleviate some of the drug’s toxicity, the improvement in overall therapeutic benefit was not significant. How to apply tumor-targeted nanomedicines to the clinic is a very worthwhile question; from new concepts and innovative research ideas in academia, to the development of pharmaceutical industry processes and technologies, to the structural cooperation between industry and academia, all affect the purpose of “laboratory to clinical”. In order to break the stagnant state of this transformation, a variety of targeting strategies have shown great potential; whether based on ligand-receptor or internal/external environment stimulation targeting strategies, they have good antitumor effect. At present, some interesting targeted drug delivery systems have been developed, such as hierarchical targeting strategies based on particle size, in which particles “from large to small” or “from small to large” can be flexibly adjusted to improve the accumulation of tumor sites; the research on the shape of particle size is no longer confined to the traditional spherical shape, and the wormlike and rod shape have shown unique advantages. A variety of new biotype NPs, such as exosomes and virus-like particles, have also been developed.

There are still many problems to be solved in order to give particles the real “active targeting ability” and make them reach the destination in a preset way. First, can the coupling method be used to stably bind and accurately expose the active sites? Second, can additional protein adsorption be effectively avoided in vivo to ensure ligand activity? Third, when the particles reach the tumor site, can they successfully bind with enough receptors to enter the cells? In order to avoid the influence of the particle itself or coupling strategy on the activity of targeted molecules, some cells have been modified by genetic engineering technology to express tumor-targeted proteins or polysaccharides on their surfaces, so as to obtain biological NPs with targeted function; or the selective adsorption of some special proteins as target molecules by the adsorption effect of proteins is used to avoid the inactivation of ligands; other scholars choose low-affinity targeting molecules to avoid binding to low-expression receptors in normal cells, and enhance the binding force between particles and high-expression receptors in tumor cells through “multivalent effect”, so as to achieve the effect of targeting tumors; however, these targeting strategies not only improve particle targeting, but also inevitably introduce new problems, such as the high requirements of preparation technology, the uncertainty of adsorbed protein and so on. It is urgent to change the targeting strategy and develop a really efficient targeted drug delivery system. Traditional targeting strategies focus on the transformation of targeted particles; changing the way of thinking and focusing on receptors may become a new research direction to improve targeted therapy. Methods such as constructing artificial targeted receptors to address the problem of variation in expression or insufficient number of receptors, or artificially placing receptor signal amplifiers at the tumor site can be used to increase the particle’s susceptibility to the tumor. The intensity of the set “receptor signal” is much greater than the specific “receptor signal” of the tumor itself, which makes the tumor site easier to be identified by the “targeted particle” and achieves the purpose of active targeting. This targeted NP drug delivery system that amplifies disease signals in vivo will have a place in the field of tumor treatment in the future.

In addition, the development of some new technologies in clinical transformation also brings new vitality to the application of targeted NPs. At present, many NPs have achieved good tumor delivery efficiency in animal experiments, but failed to achieve the desired effect in human clinical trials. To break this stagnation in clinical translation, machine learning (ML) and artificial intelligence methods were used to build a physiologically based pharmacokinetic model to help predict the absorption, distribution, metabolism and excretion properties of NPs [148]. Simulating the biodistribution of different NPs in healthy rodents and humans can effectively prevent NPs with low tumor delivery efficiency from entering preclinical trials, thereby helping researchers make decisions to reduce and improve animal studies. Secondly, ML can also establish a model to describe the complex relationship between the reaction conditions and the corresponding NPs properties and optimize the design of experiments through some developed algorithms. It is an adaptive form of experimental design that allows one to effectively identify reaction conditions and generate NPs with desirable properties [149]. At the same time, incorporating microfluidic automated synthesis technology can accelerate the identification of optimal reaction conditions and enable faster and more precise operations, such as reagent loading, mixing, heating, and on-line characterization of products, with high reproducibility, and which can be prepared on a large scale [150]. There are other technologies that have attracted widespread attention, such as the use of the particle “protein crown effect” to collect low-abundance proteins in blood circulation, and combining proteomics analysis to develop a disease protein marker detection platform; or using DNA nanotechnology to synthesize NPs with ideal size and properties through the Watson–Crick base-specific complementary pairing principle and special properties of DNA itself. The development of real targeted particles seems to be imminent; of course, many difficulties and a long industrialization process will need to be overcome to achieve these goals.

## Figures and Tables

**Figure 1 pharmaceutics-14-01919-f001:**
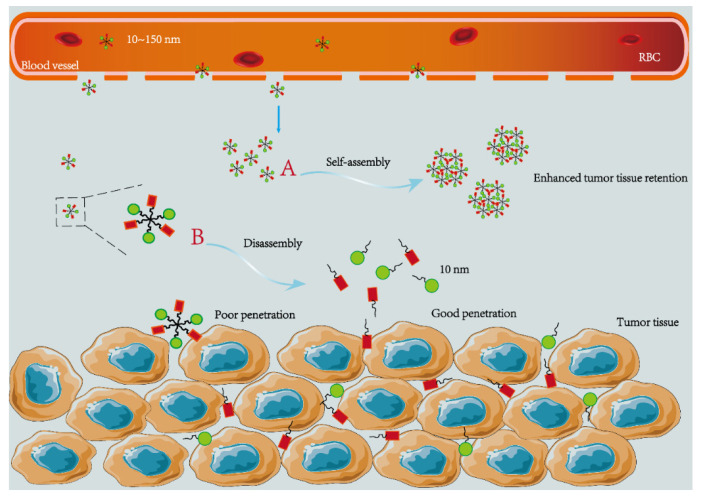
Schematic diagram of 10~150 nm particles transport in the body. The NPs consist of linkers that carry different small molecule drugs. (**A**) Self-assembly into large particles to enhance tissue retention; (**B**) rapid decomposition into small particles to improve tumor penetration.

**Figure 2 pharmaceutics-14-01919-f002:**
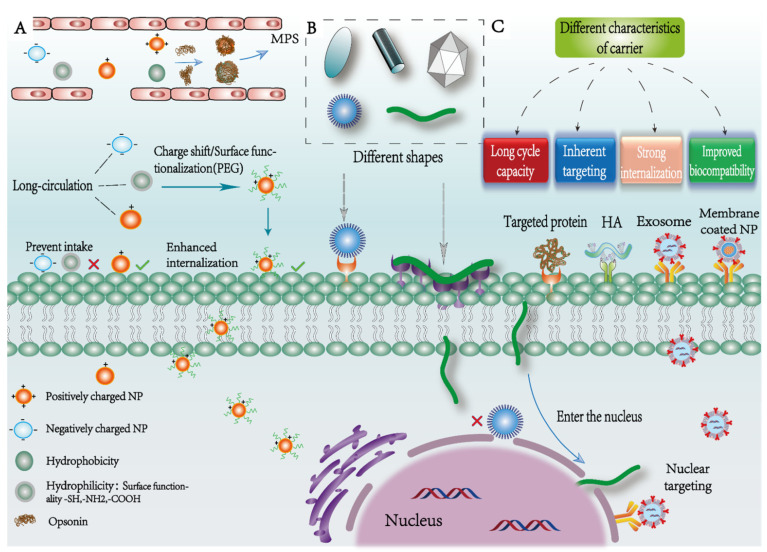
Effects of surface properties, shape and intrinsic properties on the targeting of NPs. (**A**) Transport of particles with different charge and surface chemical properties in vivo; (**B**) schematic diagram of cell uptake of NPs with different shapes; vermicular carriers find it easier to enter intracellularly by polyvalent contact with the membrane surface, and they even achieve nuclear targeting; (**C**) different intrinsic characteristics of the carrier.

**Figure 3 pharmaceutics-14-01919-f003:**
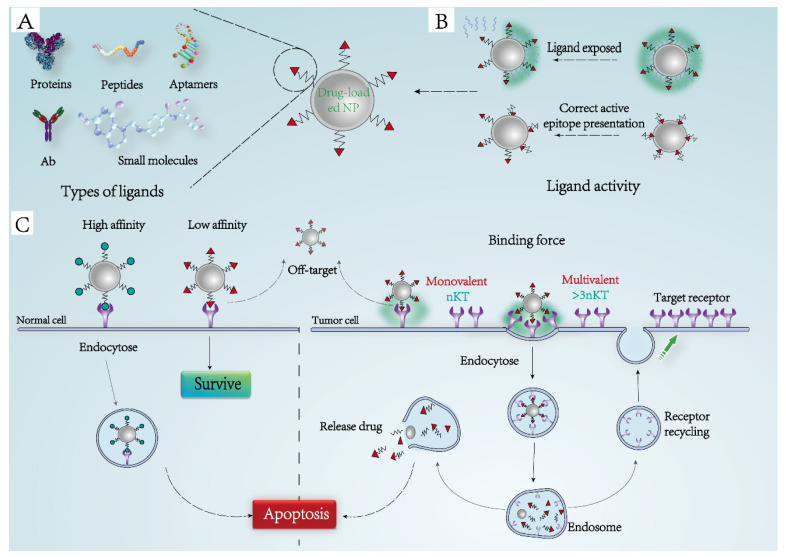
Schematic diagram of ligand-receptor interaction. (**A**) Common types of targeted ligands, including proteins (monoclonal antibodies), artificial peptides, aptamers and small molecules; (**B**) the targeting ability is demonstrated only when the ligand modified on the particle surface is exposed and exhibits the correct epitopes; (**C**) ligand-receptor binding induces endocytosis. Compared with the off-target effect of the low-affinity ligand, monovalent binding of the high-affinity ligand mediates endocytosis of drug-loaded particles and induces apoptosis. Compared with monovalent binding, the multivalent effect can significantly increase the binding force between the ligand and receptor, drive membrane invagination to form the nucleosome and reduce the number of membrane receptors, and then the receptors sprout and form individual vesicles to return to the cell surface to achieve the purpose of receptor recycling (nKT in the figure indicates the monovalent binding strength of ligand and receptor).

**Figure 4 pharmaceutics-14-01919-f004:**
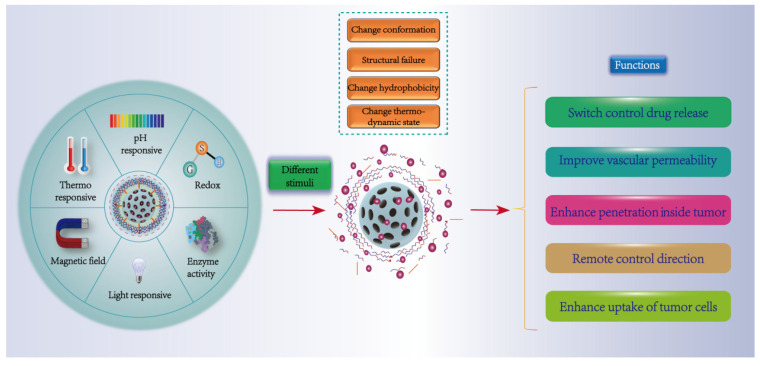
Schematic diagram of targeted drug delivery by NPs under different stimuli.

**Table 1 pharmaceutics-14-01919-t001:** List of some tumor treatments based on ligand-receptor targeting.

Types of Ligands	Samples	Aim	Result	Ref.
Proteins	Tf	Cisplatin and docetaxel were loaded into lipid hybrid NPs and modified with Tf to prepare a targeted delivery vehicle	The Tf-modified group showed stronger targeting and cytotoxicity	[100]
	EGFR antibodies	Combining EGFR antibodies with polymerized poly(lactide-coglycolide) NPs loaded with rapamycin for selective targeting of the extracellular ligand-binding domain of EGFR	MCF-7 breast cancer cells significantly augmented uptake of EGFR-coupled NPs and induced cell cycle arrest and apoptosis compared with free rapamycin and non-targeted NPs	[101]
Polysaccharides	Angiopep-2	Angiopep-2 is a complementary ligand for low-density lipoprotein-receptor-associated proteins. Coupling Angiopep-2 with paclitaxel-loaded poly(ethylene glycol)-copolymer (epsilon-caprolactone) NPs to cross the BBB and target delivery of the drug to glioma	The specific accumulation of Angiopep-2 conjugated particle complexes in the brain facilitates the nanocarrier crossing the blood-brain barrier	[102]
Artificial peptides	TGN and RGD peptides	Using virus-like particles as carriers, and selecting RGD peptides that can target tumor blood vessels and TGN, a brain-targeting peptide that can penetrate the BBB, as targeting ligands, a dual-targeted drug delivery system was developed to deliver PTX and siRNA	More effective and accurate delivery of small molecule chemotherapy drugs to the tumor site, to achieve a good anti-tumor effect	[103]
Aptamers	PSMA aptamers	The A10 RNA aptamer binds to the surface of polylactic acid NPs and encapsulates docetaxel to specifically recognize PSMA on the surface of prostate cancer cells	Significantly reduced tumor size in xenograft nude mice models of prostate cancer	[79]
Small molecules	FA	Combining PTX with pluronic123 polymer and attaching FA to the surface of the micelle by chemical coupling to obtain complex micelles	The FA-coupled polymer micelle system significantly enhanced cell uptake and anti-tumor activity, and exhibited higher anti-tumor effects and safety in animals	[104]

**Table 2 pharmaceutics-14-01919-t002:** List of some of the sensitive systems based on internal and external stimuli used for co-delivery of drugs in cancer treatments.

Stimulus Types	Carrier	Aim	Result	Ref.
Physical response (temperature, magnetism, light,)	Magnetic cationic liposomes	DPPC, DC-CHOL, DOAB, cholesterol-modified magnetic iron oxide co-delivery of DOX and SATB1-shRNA	In gastric cancer model, co-delivery of DOX and SATB1shRNA enhanced inhibition of cell growth compared with treatment alone.	[122]
	NPs	an aqueous-based formulation of glycerol-monooleate-coated magnetic NPs (GMO-MNPs) co-delivery of paclitaxel and rapamycin	High encapsulation efficiency (~95%) and drug release synergistically enhance the anti-tumor effect	[125]
	Poly-ion complex micelles (PICs)	Dendrimer phthalocyanine-encapsulated polymeric micelle (DP c/m)-mediated PCI, combined with DOX	NP-mediated double PDT/PCI effect, DOX released from endo-lysosome to the nucleus after light irradiation, improved the efficacy of PDT and PDT in the tumor	[142]
Chemical response (pH, reduction)	Hollow mesoporous silica NPs (HMSNP)	Folate-coated MSNPs bound to PEI carry DOX and siRNA	Controlled drug release and reduced the off-target action, greatly inhibiting the expression of Bcl-2 (anti-apoptotic) protein	[143]
	Amphiphilic linear-dendritic prodrugs	The amphiphilic linear dendritic vector (MPEG-B-PAMAM) was synthesized and DOX was encapsulated	The release of the drug is pH-dependent, increases cell uptake, and effectively inhibits the growth of cancer cells	[144]
	Polyplex	Star-shaped cationic polymer was prepared by γ-cyclodextrin (γ-CD) and multiple oligo-ethylenimine (OEI) arms and combined with folic acid to carry PTX	Gene transfection was enhanced and apoptosis was significant	[145]
	Polymeric micelle	ABP-PEG_3.5k_-PTX (APP) for the *co*-delivery of genes and drugs.	Improved cell uptake efficiency, low cytotoxicity	[146]
Biological response (enzyme)	Graphene oxide (GO)	Graphene oxide delivers DOX and DNA via MMP-2 cleavable PLGLAG peptide bond linked to PEI-PEG	MMP2 reacts with peptide cleavage to control drug release and enhance drug efficacy in vitro. The efficient transfection can be comparable to PEI_25k_	[140]
	Polymeric micelles	MMP2 sensitive copolymer (PEG-PP-PEI-PE) *co*-delivers siRNA and drugs	The antitumor activity of PTX and siRNA was improved	[147]

## Data Availability

Not applicable.

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
