# Peer review of "“Targeting Design” of Nanoparticles in Tumor Therapy"

_pharmaceutics, 2022, doi:10.3390/pharmaceutics14091919_

Round 1
Reviewer 1 Report
Please find the attached PDF for comments and suggestions.

Author Response
请参阅附件

Reviewer 2 Report
Comments

Reviewer 3 Report
Please find the report pdf file attached.

Reviewer 4 Report
The authors have written a good summary and informative review on nanoparticle design in Tumor Therapy. The topic is of interest, the article also showed the critical synthesis of the knowledge from the previous studies. The major context of the article not only included the findings from other reports but also provided complete discussion or suggestions that would be worthy for future research. Thus, the reviewer suggests that this article could be appropriate for this journal after minor revision.
1. Although the authors mentioned that most targeted NPs delivery strategies are still in the pre-clinical trial stage, they did not discuss what major factors that prevent NPs delivery from the clinical applications. Because only the NP design or modification targeting to resolve those factors is really meaningful for tumor therapy in the clinic.
2. Actually nanoparticles delivery strategy in cancer treatment in clinical trials has been reported, please discuss this part in this review with updated publications.
3. There are many errors for citation in the text (lines 114, 148, 180, 229, 284, 578, 581, etc.). Please check and correct them.
Round 2
Reviewer 1 Report
Dear aiuthors,
Please correct the following.
Line 505: ⍺vβ3 and ⍺vβ3 integrin --> ⍺vβ3 and ⍺vβ5 integrin
Line 137 and many other lines: Error! Reference source not found.
Reviewer 2 Report
Responses are satisfactory.